# Genetic Diversity and Population Structure of Normal Maize Germplasm Collected in South Sudan Revealed by SSR Markers

**DOI:** 10.3390/plants11202787

**Published:** 2022-10-20

**Authors:** Emmanuel Andrea Mathiang, Kyu Jin Sa, Hyeon Park, Yeon Joon Kim, Ju Kyong Lee

**Affiliations:** 1Department of Applied Plant Sciences, College of Agriculture and Life Sciences, Kangwon National University, Chuncheon 24341, Korea; 2Interdisciplinary Program in Smart Agriculture, Kangwon National University, Chuncheon 24341, Korea

**Keywords:** Maize, landrace accession, agro-ecological zone, SSR marker, genetic diversity, population structure, UPGMA dendrogram

## Abstract

Maize is one of the leading global cereals, and in South Sudan maize cultivation occurs in nearly all of the country’s agro-ecological zones. Despite its widespread cultivation, farmers in South Sudan depend on undeveloped varieties, which results in very low yields in the field. In the current study, 27 simple sequence repeat (SSR) markers were used to investigate genetic diversity and population structures among 37 landrace maize accessions collected from farmers’ fields in South Sudan. In total, 200 alleles were revealed with an average of 7.4 alleles per locus and a range from 3.0 to 13.0 alleles per locus. The observed heterozygosity values ranged from 0.06 to 0.91 with an average of 0.35. High polymorphic information content (PIC) values were identified with a mean of 0.69, which indicates the informativeness of the chosen SSR loci. Genetic structure analysis revealed a moderate genetic differentiation among the maize populations with a fixation index of 0.16, while there was very high genetic differentiation within the groups of populations of three regions with a mean fixation index (F) of 0.37. An unweighted pair group method with an arithmetic mean (UPGMA) dendrogram clustered the 37 maize accessions into three groups with 43% genetic similarity. The clustering pattern of the maize accessions was moderately consistent with their collection area. The findings of this study will provide maize breeders with a better understanding of maize diversification as well as a reserve of genetic resources for use in the selection of advantageous and useful resources for the development of maize varieties in South Sudan.

## 1. Introduction

Maize (*Zea mays* L.) is a well-known staple food and the most cultivated cereal worldwide [1]. Moreover, it has a huge economic impact as it is used in the production of biofuel, feedstock, and raw material for industry [2]. Although the introduction of maize to South Sudan is not clearly defined, it is widely believed to have been first introduced into Africa through West African countries in the 16th century, especially by Portuguese traders [3,4]. The same source states that Belgian agricultural officials discovered corn that was widely cultivated in the 18th century by the people of the Kingdom of Azande, which is in the southwest of today’s South Sudan [3]. Despite the tropical savanna climate of South Sudan, this crop is cultivated in all agricultural ecological zones, except in the semi-arid climate of the north and east [5]. Differences in climate affect local farmers in each zone, and they maintain their own landrace varieties. As a result, there is a huge amount of phenotypic variability that needs to be characterized for further use in breeding programs [6,7].

In South Sudan, maize is the second most dominant cereal after sorghum and accounts for approximately 20% of the total area of cereals. In 2021, the total yield of maize in South Sudan was 131,000 tons [8], averaging between 0.5 and 0.9 t/ha [9]. South Africa and Ethiopia are the largest maize-producing countries in Africa with average yields of 4.9 t/ha and 4.2 t/ha, respectively [10]. Compared with the yields in these countries, the maize yields per hectare in South Sudan are much lower. Although the low yield rate is associated with several challenges, the crucial challenge is the absolute dependency of South Sudanese small-scale farmers on naturally selected landraces in almost all of the field crops [11,12]. In particular, maize F_1_ hybrid commercial varieties have never been available to local farmers, while some farmers have access to open-pollinated varieties (OPV) that are being imported from neighboring countries and others that are distributed by humanitarian organizations. The yields of these imported genotypes remain extremely low owing to variations in climate. Landraces are diverse, heterogeneous populations that are commonly sought by farmers because of their adaptation to the local environment of a particular agro-ecological zone and local demand [13]. To develop varieties that are high yielding and tolerant to biotic and abiotic stresses such as drought and diseases, it is essential to understand the genetic diversity and population structure of these native genetic resources. Such an understanding is also helpful for germplasm protection and utilization for crop improvement, especially for providing breeders with the tools to evolve new and improved accessions with desired traits [11,14].

Morphological characterization is a prerequisite for breeding, but it is greatly affected by the environment. Reciprocally, molecular characterization is free from environmental effects and enables highly valuable complementary genetic information to be obtained [15,16]. Molecular markers have been extensively applied for quantitative trait loci (QTL) mapping, association studies, marker-assisted selection (MAS) for breeding and genetic research, and gene cloning [2,17]. Among all molecular markers, simple sequence repeat (SSR) markers have been considered almost the ideal markers for genetic diversity analysis [18], and their success is principally owing to a high level of polymorphism, codominance, repeatability, and reliability [16]. Therefore, SSR markers have also been optimized in maize, thus leading to a superior resolution of alleles in genetically heterozygous populations [19,20,21,22].

To our knowledge, no diversity studies based on molecular markers on South Sudanese maize landraces have been carried out to date, and there has only been very limited research regarding morphological characteristics and maize adaptation. This study is a pioneer in revealing genetic diversity and the population structure of maize landrace germplasm collected in South Sudan. A total of 15 different agricultural locations were explored in our germplasm collection survey to gather maize landrace accessions from traditional maize farms. These accessions represent different environments of maize production areas in South Sudan and were collected with the aim of understanding the genetic diversity of the indigenous accessions. For this purpose, 27 SSR primers have been constructed to study the 37 maize landrace accessions collected from South Sudan. The object of this study is to estimate the genetic diversity and population structure.

## 2. Results

### 2.1. Polymorphism of SSR Loci for Maize Landrace Accessions

To analyze the genetic diversity of maize genetic resources in South Sudan with regard to geographical conditions, we used 37 maize landrace accessions that were collected from three regions: northern (six populations), central (four populations), and southern (five populations). These maize accessions were also divided into 15 populations from the three geographic locations of South Sudan (Figure 1, Table 1). For this study, 27 SSR markers were applied to characterize the genetic diversity of the 37 maize landrace accessions collected at the 15 geographic locations across South Sudan (Table 2). For this characterization, we analyzed diverse molecular parameters including the observed number of alleles (Na), effective number of alleles (Ne), Shannon’s information index (I), observed heterozygosity (Ho), expected heterozygosity (He), major allele frequency (MAF), gene diversity (GD), and polymorphic information content (PIC) (Table 2). In total, 200 alleles were detected, segregating in the 37 maize accessions with an average of 7.4 alleles per locus, ranging from 50 to 200 bp. The Na value per locus ranged from 2.0 (for markers umc1130, umc1718, umc2075) to 6.0 (for marker umc2378), with an average of 3.6. GD value ranging from 0.574 for marker umc1380 to 0.887 for marker umc2334, with an average of 0.735, while the MAF value ranged from 0.135 for marker umc2334 to 0.622 for marker umc1380, with an average of 0.390. The Ho value ranged from 0.067 for marker umc1466 to 0.914 for marker umc1108, with an average of 0.351, while the He value ranged from 0.426 for marker umc1380 to 0.773 for marker umc2378, with an average of 0.578. The PIC value ranged from 0.532 for marker umc1024 to 0.876 for marker umc2334, with an average of 0.699. The Shannon information index value varied from 0.618 for marker umc2075 to 1.576 for marker umc2378, with an average of 1.001 (Table 2). These results show that the SSR loci that were used were effective in providing valid estimates of genetic diversity of the maize landrace populations, as represented by the mean genetic diversity indicators (I = 1.001, He = 0.578, PIC = 0.699).

### 2.2. Genetic Diversity and Genetic Differentiation among Maize Landrace Populations

The genetic diversity indices per population are presented in (Table 3). This study confirmed the Na, Ne, I, Ho, He, and F (Inbreeding coefficient or fixation index) among the 15 landrace maize populations. Na values varied from 1.160 for RE to 2.407 for MN, with an average of 1.877. Ne values varied from 1.160 for RE to 2.066 for YD, with an average of 1.678. Ho values ranged from 0.160 for RE to 0.463 for MN, with an average of 0.335. He values ranged from 0.160 for RE to 0.562 for MN, with an average of 0.413. The Shannon information index values ranged from 0.111 for RE to 0.736 for MN, with an average of 0.480. The fixation index (F) value varied from 0.000 for AW, BE, RE, and YM to 0.438 for AD, with an average of 0.167 (Table 3).

Additionally, in this study, in order to understand the genetic diversity in accordance with the climatic characteristics and geographical areas of South Sudan, we conducted a genetic diversity analysis at the regional level within the groups of populations, namely the southern region population, central region population, and northern region population (Table 4). The Na value varied from 3.000 to 3.293, with an average of 3.123, while the Ne value ranged from 2.293 to 2.364, with an average of 2.322. The Shannon information index value varied from 0.880 to 0.913, with an average of 0.899. The Ho value varied from 0.336 to 0.355, with an average of 0.348. The He value ranged from 0.552 to 0.563, with an average of 0.556. The fixation index showed a range from 0.361 to 0.3.92, with an average of 0.374 (Table 4). The analysis of the molecular variance (AMOVA) for genetic differentiation among and within the 15 populations from the three regions showed that the genetic variation that occurred among regions was only 7% (Table 5). In contrast, the remaining 93% of variability in genetic variation was represented within the regions.

### 2.3. Cluster Analysis and Population Structure

In the population structure of the 37 accessions of maize landraces from across South Sudan, it was found that the highest value of *ΔK* was K = 2 (K value describes the number of subpopulations that make up an entire population, and delta K has been suggested to help determine the most probable number of populations) for the 37 accessions of maize landraces (Figure 2). In accordance with this result, all the maize landrace accessions were divided into two main groups and one admixed group (Figure 3). At K = 2 the division was as follows: Group I included 20 maize accessions, namely AR3, WA2, TO2, WA3, YD2, TR2, AR2, AR, MN1, TO1, YD3, MN4, MN2, MD2, AD3, MD3, MN3, AD1, AD2, and AW; Group II contained 12 maize accessions, namely BO3, GO2, GO1, GO3, TR1, BO2, RE, WA1, RJ2, BE, RJ3, and YD1; and the admixed group clustered five accessions, namely RJ4, MD1, RJ1, BO1, and YM (Figure 3).

Furthermore, the dendrogram of the 37 South Sudan maize accessions revealed by an unweighted pair group method with an arithmetic mean (UPGMA) algorithm (Figure 4) showed the 37 maize accessions separated into three groups with a genetic similarity of 46.3%. Group I contained 20 maize accessions, including 12 accessions from the southern region, 3 accessions from the central region, and 4 accessions from the northern region. Group II included 16 maize accessions, including 6 accessions from the southern region, 4 accessions from the central region, and 6 accessions from the northern region. Group III contained only one maize accession, which was from the southern region.

## 3. Discussion

A comprehensive understanding of the genetic diversity of native genetic resources is crucial for initiating and managing strategies for the conservation of crop genetic resources. Despite the long history of maize cultivation in South Sudan, a lack of developed varieties still forces farmers to turn to indigenous germplasm for traditional farming. To change this prevalent pattern, the formation of knowledge about the level of genetic diversity in landraces is essential because it will allow for the more efficient and effective use of resources in plant improvement [23]. Landraces are a good source of useful alleles because they have been preferred by local farmers for their adaptation to environmental stresses [24]. In order for these genetic resources to be utilized, the characterization of native germplasms is necessary. With this aim, we used 27 SSR markers for genetic diversity and population structure estimation among maize landrace accessions, as described in the Introduction. Compared with other molecular marker systems, SSR, or microsatellite technology, has numerous advantages, viz., simple experimental methods, high reproducibility, polymorphic genetic information contents, the codominant nature of SSR polymorphisms, and abundance and distribution in plant genomes [25,26,27]. These advantages make it a worthy method for estimating genetic diversity and population structure.

In the present study, a total of 200 alleles was revealed using 27 SSR markers with an average of 7.4 alleles per locus, which is similar to or higher than the average number of alleles observed in previous studies using SSR markers [21,28,29]. This high average of alleles per locus can be attributed to high genetic diversity in the investigated genotypes. The PIC value affords a fairer estimation of diversity than the actual number of alleles because it takes into account the relative frequencies of each allele present [23,30]. In our study, the overall average PIC for the SSR loci found was 0.699, which is in agreement with the PIC of 0.69 observed in the Japanese inbred lines of maize accessions [31]. Compared with our findings, Thakur et al. (0.43) [32] and Belalia (0.57) [16] obtained lower values; a higher PIC value was recorded in a Turkish landrace maize population (0.72) using SSR markers [30]. These results indicate that the SSR markers used in this study provided adequate information for estimating the level of genetic diversity in South Sudan’s maize landraces. In particular, some SSR primers, such as umc1108, umc1316, umc1607, umc2275, and umc2378, which showed a high number of alleles and high values of genetic diversity, were considered as useful markers for evaluating the genetic diversity among South Sudan maize accessions. Landraces of maize accessions from Algeria and Northwest Argentina were found to have similar overall genetic diversity values (He = 0.57) as found in this study (He = 0.57) [7,33]. In contrast, Noldin et al. [34] reported a slightly lower genetic diversity (He = 0.48) in Paraguayan maize accessions than the average reported by this and some other reports (He = 0.57). Conversely, few studies have stated higher genetic diversity than the present study [16,22,35,36] with the highest level obtained by Yao et al. [37] (He = 0.7) for maize landraces from the Wuling Mountains region in China. In addition, the apparent difference between the observed and the expected heterozygosity values in this study reflects a deficiency in heterozygosity, which may be the result of low cross-fertility and high self-fertilization rates [38,39]. Another possible reason could be that small-scale farmers usually change the relevant landrace each year by extracting seeds from a small number of ears [28]. However, matching results in heterozygosity deficiency were registered in several previous studies [16,28,34].

The fixation index (F) calculates population differences by virtue of the genetic structure, and a value over 0.15 can be considered significant in differentiating populations [40,41]. In our study, the mean coefficient of genetic differentiation between accessions was identified as F = 0.16 (Table 3). This result is in accordance with the result found in maize landraces from Mexico by Pineda-Hidalgo et al. (2013) [42] as well as that recorded in maize landrace accessions of the Ivory Coast in 2016 [43]. Furthermore, the F index mean within the groups of populations from the three regions of South Sudan (southern region, central region, and northern region) was F = 0.37 (Table 4). This finding is higher than that that was found in Saharan Algerian maize populations (F = 0.22) in 2018 [16] and in accordance with the F mean (F = 0.36) reported in Indian maize landrace populations in 2013 [44]. This result shows a greater genetic differentiation within the groups of populations from the three regions of South Sudan. Moreover, the F value for the northern region populations (0.39) is slightly higher than the F value of the other two regions. This indicates a higher genetic variability. Although in general the F values of all three regions are contiguous, the observed standard of differentiation suggests a moderate gene flow between the populations, which could be attributed to the effect of pollen dispersal and seed mixing/exchange between the neighboring populations [16,44] because it is known that local farmers exchange seeds among themselves in order to raise crop productivity [41,45]. Furthermore, the centers of environmental diversity where maize is being cultivated as a rainfed crop are heterogenous. Usually, the first planting is performed in April in the southern region (equatorial climate, high rainfall, long growing period), while in the central and northern regions planting for the first season starts later in May and June. Consequently, the second growing season, especially in the southern region, begins in September up to late October. This varied condition might have influenced the level of molecular differentiation of local landrace maize accessions. In addition, AMOVA is a satisfactory grouping criterion for evaluating the variation within and among populations. In accordance with the result of the inbreeding coefficient value (F), the AMOVA revealed a greater level of genetic variation within rather than among regions (Table 5), which matches the results recorded in previous studies [32,46,47]. According to Da Silva et al., outcrossing species ordinarily sustain the genetic variation within populations while genetic variation is lower among populations [47].

Furthermore, to have a clear understanding of the information of genetic diversity and population structure of the 37 maize landrace accessions collected from South Sudan, we used the following two statistical methods: a model-based approach with STRUCTURE software and a UPGMA dendrogram with NTSYS-pc V2.1. The results of STRUCTURE revealed that the 37 maize landrace accessions could be split into two main groups and one admixed group based on the *∆K* value (Figure 3). At *K* = 2, Group I consisted of 20 maize accessions representing 54.05% of the total genotypes under study, and most accessions assigned in this group were derived from central region populations, while the southern and northern region populations were represented equally. Group II comprised 12 maize accessions representing 32.43% of the total tested genotypes, and most accessions positioned in this group were derived only from the southern and northern region populations. The admixed group included five accessions from the southern and central region populations only (Figure 3).

Moreover, the UPGMA dendrogram results based on SSR marker data classified the 37 South Sudan maize landrace accessions into three groups with 43% genetic similarity. The major groups I and II occupied almost all the genotypes, while group III had only one accession from a different origin (Figure 4). The clustering was in accordance with the agro-ecological climate of South Sudan, where there are high rainfall and humid tropical savannahs in the southern and central regions of the country and a subtropical climate with semi-arid lands in some parts of the northern region. Group I involved 20 accessions and further separated into three subgroups; 11 accessions in this group derived from populations representing southern parts of South Sudan, while 5 accessions derived from populations representing the central areas and 3 accessions derived from populations from northern areas. Group II consisted of 16 accessions with two minor subgroups; 6 accessions were derived from central region populations, while 6 accessions were from northern region populations and 4 accessions were from southern region populations. Group III revealed one accession, MN2, which is an outlying accession representing a unique genotype derived from the southern part of the country. Most of the accessions were positioned as expected. The placing of accessions from different collection areas in the same group indicates a close genetic association regardless of their diverse origin. The findings suggest that the landrace genotypes of maize may be frequently exchanged among the regions by farmers and that this might have occurred through different routes corresponds clearly with the hypothesis of seed mixing/exchanging and seed trade between small-scale farmers [32].

Even though this paper remains a preliminary study for understanding South Sudan maize germplasm diversity, these findings will be very useful in forming standards for the selection of more genetic materials and allelic germplasm resources with substantial genetic diversity for breeding programs. In this study, genetic diversity analysis was performed using SSR markers on native maize accessions collected in South Sudan, and the results are expected to provide useful and interesting information on the conservation of landrace maize genetic resources and the selection of useful resources for maize breeding programs in South Sudan.

## 4. Materials and Methods

### 4.1. Plant Materials and DNA Extraction

We studied 37 maize landrace accessions that were collected from populations found in 15 geographic locations of South Sudan. The 37 maize accessions were divided into three populations in relation to geographical conditions, namely northern, central, and southern regions in South Sudan (Figure 1, Table 1). The southern region landrace accessions were collected from the following locations: Bor, Rejaf, Mangalla, Gondokoro, Torit, and Yambio. The central region landrace accessions were collected from Aber, Madhok, Adull, and Tonj, while the northern region landrace accessions were collected from Wau, Aweil, Yida, Renk, and Bentiu. In the southern region, 17 maize accessions were collected from Bor (3 accessions), Rejaf (4 accessions), Mangalla (4 accessions), Gondokor (3 accessions), Torit (2 accessions), and Yambio (1 accessions); in the central region, 11 maize accessions were collected from Aber (3 accessions), Madhok (3 accessions), Adull (3 accessions), and Tonj (2 accessions); and in the northern region 9 maize accessions were collected from Wau (3 accessions), Aweil (1 accession), Yida (3 accessions), Renk (1 accession), and Bentiu (1 accession). The genomic DNA of young leaf tissues was extracted in accordance with the Dellaporta et al. (1983) [48] method with minor modifications.

### 4.2. SSR Analysis and DNA Electrophoresis

In this study, a total of 100 SSR primer sets (distributed over 10 maize chromosomes) were used in a preliminary experiment using six landrace accessions collected from three populations (southern, central, and northern) of South Sudan. The 27 SSR primer sets, as shown in Table 2, were selected and used for genetic diversity analysis, because they showed different amplification patterns and good polymorphisms among the six landrace accessions. The SSR primer sets used in this study were derived from MaizeGDB (http://www.maizegdb.org/, accessed on 1 September 2021). SSR amplification was performed using EX *Taq* PCR kit (Takara, Ohtsu, Japan). A total volume of 20 µL of product was composed of 20 ng genomic DNA, 0.2 mM dNTPs, 0.5 µM forward and reverse primers, 1×EX *Taq* buffer, and 1 unit of EX *Taq* polymerase for polymerase chain reaction (PCR) of the SSR loci. The PCR protocol was conducted as follows: first step, initial denaturation at 94 °C for 5 min; second step, 1 min of denaturation at 94 °C, 1 min of annealing at 65 °C, and 2 min of extension at 72 °C. The second step was repeated 36 times with the temperature of the annealing stage lowered by decreases of 1 °C at every annealing stage until an eventual annealing temperature of 55 °C was reached. After completing the first two steps, a final third step was carried out of 5 min of extension at 72 °C. DNA electrophoresis analysis was conducted with a mini vertical electrophoresis system (MGV-202-33, CBS Scientific Company, San Diego, CA, USA) for the PCR products. Three µL of the final product was mixed with 3 µL of formamide loading dye (98% formamide, 0.02% xylene C, 0.02% BPH, and 5 mM NaOH). Two µL from each sample was loaded onto a 6% acrylamide-bis acrylamide gel (19:1) in 0.5X Tris-borate-EDTA (TBE) buffer and electrophoresed at 250 V for 30–40 min. The separated fragments were then visualized using ethidium bromide (EtBr).

### 4.3. Data Analysis

DNA fragments amplified using the SSR primers were scored as present (1) or absent (0). The number of alleles, PIC, MAF, and GD were identified using Powermarker version 3.25 [49]. Popgen32 V1.02 [50] was applied to obtain information on the Na, Ne, Ho, He, and I. GS between each pair of accessions were calculated using the Dice similarity index [51]. A similarity matrix was then used to construct a dendrogram, adopting the UPGMA via the application of SAHN-Clustering from NTSYS-pc V2.1 [52]. The population structure among the 37 maize accessions was investigated by using STRUCTURE 2.2 software [53]. Five independent runs were proceeded with K values ranging from 1 to 10, with 100,000 cycles for both burn-in and run length. The delta *K* statistic, based on the degree of change in the log probability of data between K values [54], was calculated with STRUCTURE HARVESTER (http://taylor0.biology.ucla.edu/structHarvester/, accessed on 25 February 2022) based on the STRUCTURE results. AMOVA was performed using GenAlEx 6.5 software [55]. Fixation index (F, inbreeding coefficient) was calculated using the following formula [56]
(1)$$F=1-\frac{Ho}{He}$$

## 5. Conclusions

This study is a pioneering approach to the evaluation of genetic diversity and population structure of South Sudan maize. A wide understanding of genetic relatedness is crucial for the improvement of any crop. In order to carry out this study, we used 27 SSR markers to investigate the genetic diversity and population structure of native maize accessions collected from across South Sudan. The amplification of polymorphic loci disclosed the potential of the selected SSR markers for investigating genetic diversity among the collected genotypes. Generally, the indices revealed distinct molecular diversity among the maize landrace populations. In addition, the overall genetic index of inbreeding coefficient (Fst) showed a moderate genetic differentiation among the maize populations and a huge genetic differentiation within the groups of populations from the three regions (southern, central, and northern populations). AMOVA revealed greater genetic variation within the populations rather than among the regions. Moreover, although the results of population structure and UPGMA analysis based on the molecular data of the SSR markers appeared to be mainly in accordance with the geographic location, a number of accessions from different collection areas were positioned in the same group. This result suggests that the mixing of landrace genotypes of South Sudan maize across the regions by local farmers might have occurred in different ways. Even though this paper remains a preliminary study for understanding the diversity of maize in South Sudan, these findings will be very useful in forming standards for genetic materials and for the selection of allelic resources with substantial genetic diversity for breeding programs.

## Figures and Tables

**Figure 1 plants-11-02787-f001:**
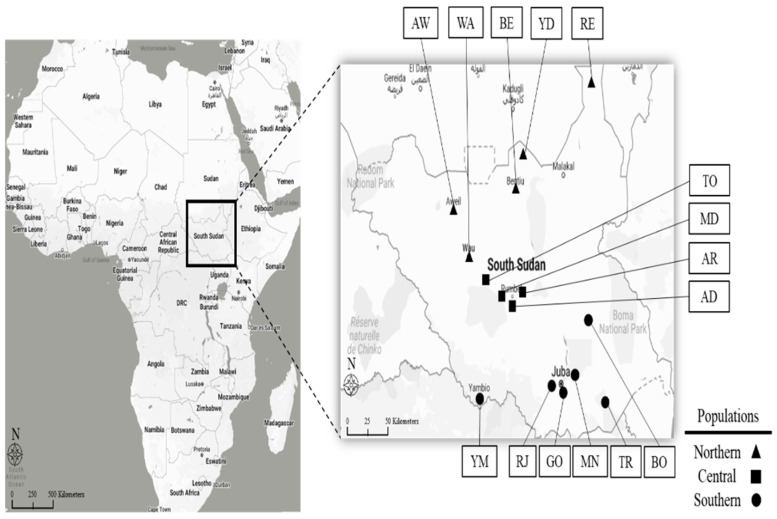
Maps showing geographic location of collection site of maize landrace populations sampled in South Sudan (●: maize population of southern region, ■: maize population of central region, and ▲: maize population of northern region). Refer to Table 1, for the full name of locations’ abbreviations used in the Figure 1.

**Figure 2 plants-11-02787-f002:**
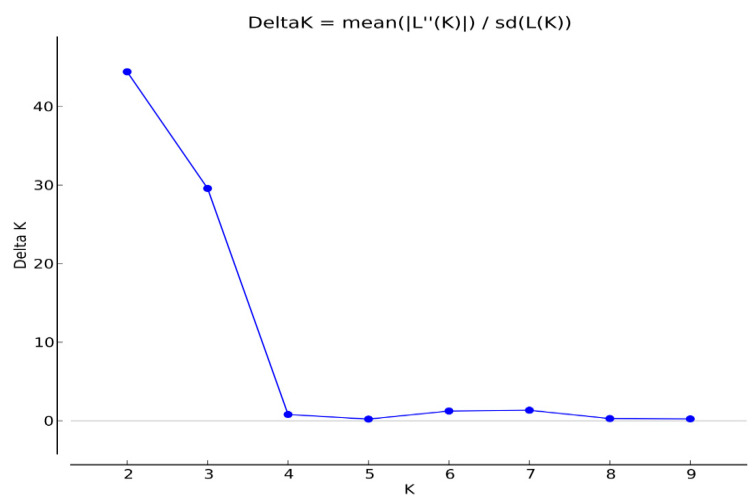
The magnitude of ∆K as a function of K; the peak value of ∆K was at K = 2, suggesting two genetic groups in the 37 maize landrace accessions.

**Figure 3 plants-11-02787-f003:**
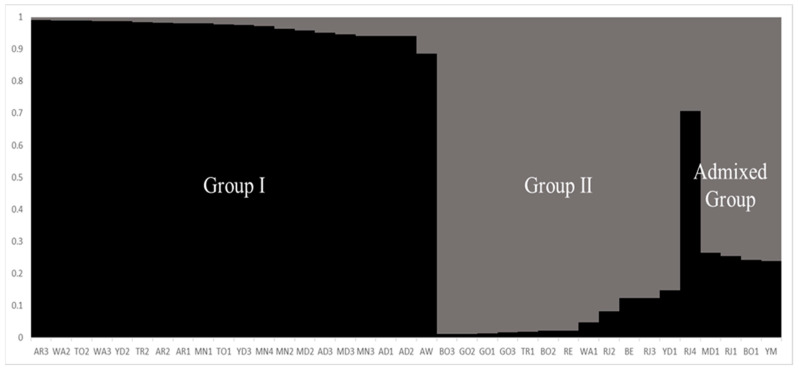
The population structure pattern for the highest ΔK value, K = 2, of the 37 maize landrace accessions.

**Figure 4 plants-11-02787-f004:**
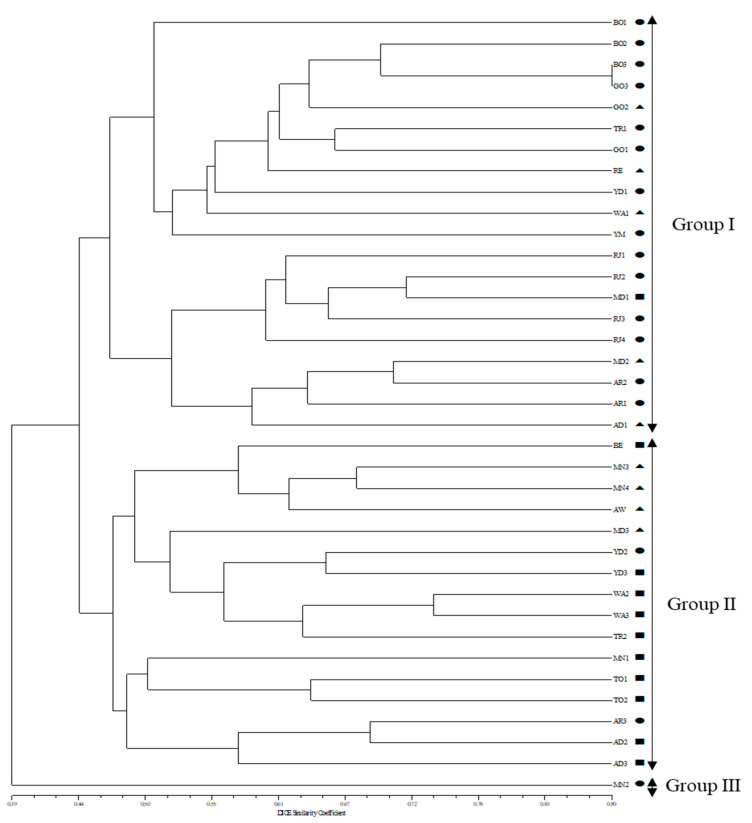
UPGMA dendrogram of 37 maize accessions based on 27 SSR markers (●: Southern region maize accessions, ▲: Northern regions maize accessions, ■: Central region maize accessions).

**Table 1 plants-11-02787-t001:** Summary of 37 maize landrace accessions collected from various locations in South Sudan.

Population	Reference Name (Abbr)	Field-Plot Name	Field-Plot Location	No. Genotype Sampled	Biological Status	Geographical Coordinates	
						Latitude	Longitude
Southern populations	**BO**	Bor	Bor, Gonglei State	3	Landrace	6°12′47.8″	31°33′56.0″
	**RJ**	Rejaf	Central Equatoria State	4	Landrace	4°45′07.9″	31°34′18.8″
	**MN**	Mangalla	Central Equatoria State	4	Landrace	5°10′48.0″	31°46′04.8″
	**GO**	Gondokor	Central Equatoria State	3	Landrace	4°54′00.0″	31°40′00.1″
	**TR**	Torit	Torit, Magwi, Eastern Equatoria	2	Landrace	4°24′36.7″	32°34′26.4″
	**YM**	Yambio	Western Equatoria State	1	Landrace	4°34′39.4″	28°23′55.7″
Central populations	**AR**	Aber	Rumbek, Lake State	3	Landrace	6°48′25.9″	29°40′44.0″
	**MD**	Madhok	Rumbek, Lake State	3	Landrace	6°42′22.7″	29°40′46.6″
	**AD**	Adull	Rumbek, Lake State	3	Landrace	6°37′00.1″	29°57′00.0″
	**TO**	Tonj	Tonj, Kuajok, warrap State	2	Landrace	8°18′14.4″	27°59′36.2″
Northern populations	**WA**	Wau	Wau, Raja, western bahr ghazhal State	3	Landrace	7°42′33.1″	27°59′00.6″
	**AW**	Aweil	Northern Bahr el Ghazal State	1	Landrace	8°46′01.6″	27°23′59.3″
	**YD**	Yida	Ruweng Administrative Area	3	Landrace	10°06′13.0″	30°05′25.1″
	**RE**	Renk	Renk, Upper Nile State	1	Landrace	11°44′34.8″	32°48′16.1″
	**BE**	Bentiu	Bentiu, Unity State	1	Landrace	9°13′53.4″	29°48′01.8″
**Total**				**37**			

**Table 2 plants-11-02787-t002:** Genetic parameters summary of 27 SSR markers used for analyzing the 37 maize landrace accessions.

SSR Loci Name	Chr.	Allele Size	Sample Size	Allele No	Na	Ne	I	Ho	He	MAF	GD	PIC
bnlg1605	9	100–150	74	8	5	1.744	0.883	0.351	0.432	0.595	0.614	0.590
umc1024	1	102–150	70	6	3	1.895	0.785	0.086	0.479	0.595	0.580	0.532
umc1066	7	110–150	72	6	3	1.882	0.762	0.389	0.475	0.459	0.652	0.590
umc1082	8	150–200	66	9	4	3.571	1.326	0.485	0.731	0.243	0.859	0.844
umc1101	4	110–150	38	6	3	2.431	0.972	0.105	0.605	0.486	0.673	0.627
umc1108	8	100–130	70	13	4	3.695	1.342	0.914	0.740	0.378	0.808	0.792
umc1130	8	100–130	74	3	2	1.810	0.640	0.351	0.454	0.486	0.614	0.536
umc1175	7	60–80	66	7	4	3.081	1.224	0.273	0.686	0.324	0.790	0.762
umc1227	5	60–110	56	8	4	3.045	1.218	0.214	0.684	0.243	0.824	0.800
umc1303	3	90–120	72	5	3	1.861	0.704	0.417	0.469	0.432	0.650	0.583
umc1315	4	90–120	74	4	3	1.938	0.809	0.568	0.491	0.432	0.614	0.534
umc1316	8	95–130	72	11	4	3.011	1.205	0.389	0.677	0.324	0.831	0.815
umc1339	2	110–150	52	9	4	3.108	1.220	0.192	0.692	0.297	0.801	0.774
umc1380	7	130–150	70	6	3	1.724	0.724	0.143	0.426	0.622	0.574	0.543
umc1454	4	130–160	68	7	3	1.959	0.852	0.412	0.497	0.459	0.723	0.693
umc1466	2	150–200	60	5	3	1.744	0.765	0.067	0.434	0.568	0.621	0.582
umc1607	7	100–140	70	11	4	2.162	1.014	0.314	0.545	0.324	0.830	0.813
umc1718	4	150–200	70	4	2	1.994	0.692	0.257	0.506	0.378	0.690	0.628
umc1872	4	100–130	74	7	4	3.009	1.208	0.432	0.677	0.297	0.811	0.786
umc2075	2	140–200	68	4	2	1.745	0.618	0.265	0.433	0.514	0.644	0.591
umc2135	8	150–200	52	7	4	2.136	0.887	0.231	0.542	0.351	0.754	0.717
umc2275	4	130–200	74	11	5	3.423	1.410	0.865	0.718	0.270	0.840	0.822
umc2286	2	70–110	56	6	3	2.085	0.892	0.214	0.530	0.405	0.741	0.704
umc2329	10	150–200	66	10	4	2.447	1.036	0.182	0.601	0.297	0.830	0.810
umc2334	7	50–100	74	10	4	3.695	1.341	0.595	0.739	0.135	0.887	0.876
umc2378	3	90–130	62	12	6	4.178	1.576	0.548	0.773	0.216	0.874	0.861
umc2540	7	140–200	64	5	3	2.293	0.931	0.219	0.573	0.405	0.717	0.671
Max			74	13	6	4.178	1.576	0.914	0.773	0.622	0.887	0.876
Min			38	3	2	1.724	0.618	0.067	0.426	0.135	0.574	0.532
Mean			66.1	7.4	3.6	2.506	1.001	0.351	0.578	0.390	0.735	0.699
Total			1784	200								

Note: Na—Number of different alleles; Ne—Number of effective alleles; I—Shannon’s Information Index; Ho—observed Heterozygosity; He—Expected Heterozygosit; MAF—major allele frequency; GD—genetic diversity; PIC—polymorphic information contents.

**Table 3 plants-11-02787-t003:** Genetic diversity parameters of 15 maize landrace populations characterized with 27 SSR markers.

Geographic Population	Sample Size	Na	Ne	I	Ho	He	F
AD	5	2.039	1.778	0.583	0.263	0.468	0.438
AR	5	2.148	1.925	0.601	0.395	0.457	0.135
AW	2	1.240	1.240	0.166	0.240	0.240	0.000
BE	2	1.192	1.192	0.133	0.192	0.192	0.000
BO	5	2.148	1.790	0.593	0.444	0.474	0.063
GO	5	1.852	1.610	0.476	0.259	0.383	0.322
MD	6	2.185	1.893	0.620	0.364	0.473	0.230
MN	7	2.407	2.058	0.736	0.463	0.562	0.176
RE	2	1.160	1.160	0.111	0.160	0.160	0.000
RJ	7	2.077	1.730	0.524	0.292	0.373	0.219
TO	4	1.800	1.677	0.495	0.400	0.467	0.143
TR	4	1.889	1.770	0.525	0.296	0.488	0.392
WA	6	2.185	1.846	0.612	0.377	0.475	0.208
YD	6	2.407	2.066	0.726	0.457	0.553	0.174
YM	2	1.429	1.429	0.297	0.429	0.429	0.000
Max	7	2.407	2.066	0.736	0.463	0.562	0.438
Min	2	1.160	1.160	0.111	0.160	0.160	0.000
Mean	4.5	1.877	1.678	0.480	0.335	0.413	0.167
Total	68						

Note: Na—Number of different alleles; Ne—Number of effective alleles; I—Shannon’s Information Index; Ho—observed Heterozygosity; He—Expected Heterozygosity; F—Fixation Index.

**Table 4 plants-11-02787-t004:** Results of microsatellite analysis for group of maize populations per region.

Population	No. Populations	Sample Size	Na	Ne	I	Ho	He	F
Southern populations	6	30	3.296	2.293	0.913	0.353	0.553	0.361
Central populations	4	20	3.074	2.364	0.903	0.355	0.563	0.369
Northern populations	5	17	3.000	2.310	0.880	0.336	0.552	0.392
Mean		22.3	3.123	2.322	0.899	0.348	0.556	0.374
Total		67						

Note: Na—Number of different alleles; Ne—Number of effective alleles; I—Shannon’s Information Index; Ho—observed Heterozygosity; He—Expected Heterozygosity; F—Fixation Index.

**Table 5 plants-11-02787-t005:** Analysis of molecular variance (AMOVA) based on SSR marker in maize landrace populations.

Source	df	SS	MS	Est. Var.	%
Among Regions	2	63.780	31.890	1.282	7%
Within Regions	34	567.193	16.682	16.682	93%
Total	36	630.973		17.964	100%

Note: df—Degrees of freedom; SS—Sum of squares; MS—Mean of squares; Est. Var. —Estimate of variance; %—Percentage of total variation.

## Data Availability

Data is contained within the article.

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
