# Peer review of "Genetic Diversity and Population Structure of Normal Maize Germplasm Collected in South Sudan Revealed by SSR Markers"

_plants, 2022, doi:10.3390/plants11202787_

Round 1

Reviewer 1 Report

This manuscript used SSR to esstimate the genetic diversity and population structure of the maize landrace accessions collected from South Sudan. The information from this study is useful for maize research in germplasm distribution worldwide, and for further breeding work in the country. After reviewing, I suggested this paper could be accepted for publication in Plants, but need substantial revisions. The comments or suggestions are as below:

1: How the authors designed and ensured the suitability of the 27 SSR markers. These details need to be addressed in the part of materials and methods

2: Did the authors find the synonyms of the accessions? Because from the result of Figure 3. UPGMA, there are some accessions with close genetic association.

3: Did the authors do the phenotyping of the accessions?

Author Response

1: How the authors designed and ensured the suitability of the 27 SSR markers. These details need to be addressed in the part of materials and methods.

-> In new version manuscript, we explained “how the 27 SSR markers designed and ensured suitability as follows: “In this study, a total of 100 SSR primer sets (distributed over 10 maize chromosomes) were used in a preliminary experiment using six landrace accessions collected from three populations (southern, central and northern) of South Sudan. The 27 SSR primer sets, as shown in Table 2, were selected and used for genetic diversity analysis, because they showed different amplification patterns and good polymorphisms among the six landrace accessions. The SSR primer sets used in this study were derived from MaizeGDB (http://www.maizegdb.org/, accessed on 1 september 2021)..”  See L. 320-326, in new version manuscript.

2: Did the authors find the synonyms of the accessions? Because from the result of Figure 3. UPGMA, there are some accessions with close genetic association.

-> Yes, our results showed that the close genetic relationships that emerge between some accessions from different regions of the collection can result in some degree of genotypic admixture. Therefore, in the discussion section part (L. 273-293) and the conclusion section part (L. 371-376), we explained the reasons for some accessions with close genetic relationships.

3: Did the authors do the phenotyping of the accessions?

Our study is to evaluate the genetic diversity of native maize genetic resources collected in South Sudan. For reminder, unfortunately, this research work was performed in South Korea and was not successful in phenotypic characterization due to climate differences between South Korea and South Sudan. Although morphological characteristics are useful for diversity evaluation, the SSR markers used in this study are considered sufficient for diversity evaluation of the native maize genetic resources collected in South Sudan.

Reviewer 2 Report

The manuscript in reference describes the genetic diversity and population structure of normal maize germplasm collected in South Sudan. The manuscript has relevant information and results that will be interesting for readers. However, some issues should be addressed prior to further consideration.

·         Ln 14:   Preposition usage: Consider changing 'at the field' to 'in the field'

·         Ln 15:   Should identify abbreviations of SSR upon the first usage

·         Ln 38:   Consider removing the verb 'being'.

·         Ln 48-50:  The phrase 'two largest producers of South Africa and Ethopia' gives a perception that 'South Africa' and 'Ethopia' are the products. Consider rephrasing the sentence.

·         Ln 50-52: The phrase 'and a serious....in the country' appears non-sequitur in this sentence. Paraphrase the sentence.

·         Ln 95: The order of Tables and Figures should be in proper order. Section 2.1 first mention Figure 4 and Table 5. The first mention should go to figure 1 and so on. Therefore, reorder your figures and tables accordingly. For example, Figure 4 should come first and be labeled as Figure 1.

·         Ln 112: Table 1 needs to be reordered and placed accordingly.

·         Ln 120 & 146: The fixation index is represented as F in line 120, whereas in line 146, F is abbreviated as 'Fixation index'. Although the inbreeding coefficient 'F' stands for the Fixation index, I would suggest mentioning both terms in line 120.

·         Ln 151: Define delta K before using it for the first time.

·         Ln 165-175: Authors should explain why the UPGMA algorithm was selected as the method for dendrogram construction. Did the author compare the other dendrogram construction algorithms such as NJ by correlation analysis (Please refer to https://pubmed.ncbi.nlm.nih.gov/32559639/).

·         Ln 222: The fixation index is represented as 'Fst', while in line 146 as 'F'. Keep it uniform.

·         Ln 248: The 'inbreeding coefficient value is represented as 'Fst', while in line 120 as 'F'. Keep it uniform.

·         Table 5: Format the font type and size of the first column and keep them uniform with the rest of the columns.

·         Ln 352: Correct the verb form for 'pioneer'.

·         Ln 352: Correct the preposition 'on' to 'to'

·         Ln 367: Remove the article before the word 'mixing'

Author Response

Ln 14:   Preposition usage: Consider changing 'at the field' to 'in the field'

-> Based on the reviewer’s comments, we changed 'at the field' to 'in the field'.  See L. 14, in new version manuscript.

Ln 15:   Should identify abbreviations of SSR upon the first usage.

-> We corrected SSR abbreviation to the full name as: simple sequence repeat (SSR).  See L. 15, in new version manuscript.

Ln 38:   Consider removing the verb 'being'.

-> Based on the reviewer’s comments, we removed the verb 'being'.  See L. 39, in new version manuscript.

Ln 48-50:  The phrase 'two largest producers of South Africa and Ethopia' gives a perception that 'South Africa' and 'Ethopia' are the products. Consider rephrasing the sentence.

Ln 50-52: The phrase 'and a serious....in the country' appears non-sequitur in this sentence. Paraphrase the sentence.

-> We changed the sentence “Ln 48-50 and Ln 50-52” in old manuscript as follow: “South Africa and Ethiopia are the largest maize-producing countries in Africa with average yields of 4.9 t/ha and 4.2 t/ha, respectively [10]. Compared with the yields in these countries, the maize yields per hectare in South Sudan are much lower. Although the low yield rate is associated with several challenges, the crucial challenge is the absolute dependency of South Sudanese small-scale farmers on naturally selected landraces in almost all of the field crops [11,12].”  See L. 48-54, in new version manuscript.

Ln 95:  The order of Tables and Figures should be in proper order. Section 2.1 first mention Figure 4 and Table 5. The first mention should go to figure 1 and so on. Therefore, reorder your figures and tables accordingly. For example, Figure 4 should come first and be labeled as Figure 1.

-> Based on the reviewer’s comments, we changed the order of Tables and Figures, as (Figure 4, Table 5) to (Figure 1, Table 1) and placed accordingly.  See L. 94, 116, 119-121, in new version manuscript.

Ln 112:  Table 1 needs to be reordered and placed accordingly.

-> We changed ‘Table 1’ to ‘Table 2’ and placed accordingly.  See L. 122, in new version manuscript.

In addition, the rest of ‘the figures and tables’ were modified according to the order.  See L. 125, 133, 134, 139-140, 145, 147, 150, 152, 159, 160, 165, 168, 171, 173, 181, 232, 236, 256, 265, 272, 276, 307, in new version manuscript.

Ln 120 & 146:  The fixation index is represented as F in line 120, whereas in line 146, F is abbreviated as 'Fixation index'. Although the inbreeding coefficient 'F' stands for the Fixation index, I would suggest mentioning both terms in line 120.

-> Based on the reviewer’s comments, we mentioned both terms for ‘F’ like as (Inbreeding coefficient or fixation index) in line 126.  See L. 126, 356, in new version manuscript.

Ln 151:  Define delta K before using it for the first time.

-> Based on the reviewer’s comments, we define delta K as follow: ----the highest value of ΔK was K= 2 (K value describes the number of subpopulation that make up the entire populations, and ΔK has been suggested to help determining the most probable number of populations) for the 37 accessions of maize landraces (Figure 2).  See L. 156-158, in new version manuscript.

Ln 165-175:  Authors should explain why the UPGMA algorithm was selected as the method for dendrogram construction. Did the author compare the other dendrogram construction algorithms such as NJ by correlation analysis (Please refer to https://pubmed.ncbi.nlm.nih.gov/32559639/).

-> This study was conducted to investigate the genetic diversity of maize, traditionally cultivated in South Sudan.

The method of dendrogram construction based on similarity for these maize native resources was first analyzed and compared using both UPGMA and NJ algorithms. In general, the dendrogram construction using the NJ algorithm is used as a useful method to understand the evolution and differentiation process of plant species, and the phylogenetic differentiation process within and between species. However, NJ algorithm methods has some limitation such as resulting tree strongly depends on the model of evolution used, also it has undesirable feature that it usually assigns negative lengths to some of the branches. In our study, we compared the results of both clustering methods, and found that result of clustering using UPGMA was better than NJ (data not shown). Therefore, the result of genetic distance based on similarity in UPGMA is preferable and better in explaining relationships, and was also in accordance with the structure result, that's why we selected it.

Ln 222:  The fixation index is represented as 'Fst', while in line 146 as 'F'. Keep it uniform.

-> Based on the reviewer’s comments, we changed 'Fst' to 'F'.  See L. 229, in new version manuscript. 

Also, in the rest of the sentences, Fst was changed to F.  See L. 23, 238, 240, 241, 242, in new version manuscript

Ln 248:  The 'inbreeding coefficient value is represented as 'Fst', while in line 120 as 'F'. Keep it uniform.

-> Based on the reviewer’s comments, we changed 'Fst' to 'F'.  See L. 255, in new version manuscript. 

Table 5:  Format the font type and size of the first column and keep them uniform with the rest of the columns.

-> We changed Table 5 to Table 1, and also formatted the font type and size of the first column and keep them uniform with the rest of the columns.  See Table 1, L. 116, in new version manuscript.

Ln 352:  Correct the verb form for 'pioneer'.

-> We corrected the verb form of 'pioneer' to ‘pioneering’.  See L. 360, in new version manuscript.

Ln 352: Correct the preposition 'on' to 'to'

-> We corrected the preposition 'on' to 'to'.  See L. 360, in new version manuscript.

Ln 367: Remove the article before the word 'mixing'

-> We removed the article before the word 'mixing'.  See L. 375, in new version manuscript.

Round 2

Reviewer 1 Report

After reviewing, I think the major problems that I concern previously have been clarified in the revised manuscript. I suggest that the revised version of the manuscript can be accepted for the Journal 'Plants'.